# MMM and MMMSynth: Clustering of heterogeneous tabular data, and synthetic data generation

Chandrani Kumari[1,2], Rahul Siddharthan[1,2]*

**1** The Institute of Mathematical Sciences, Chennai, India, **2** Homi Bhabha National Institute, Mumbai, India

* rsidd@imsc.res.in

**Data Availability Statement:** Data are from public sources cited in the manuscript. The software described, MMM and MMMsynth are available on \href{https://github.com/rsidd120/MadrasMixtureModel}{https://github.com/

## Abstract

We provide new algorithms for two tasks relating to heterogeneous tabular datasets: clustering, and synthetic data generation. Tabular datasets typically consist of heterogeneous data types (numerical, ordinal, categorical) in columns, but may also have hidden cluster structure in their rows: for example, they may be drawn from heterogeneous (geographical, socioeconomic, methodological) sources, such that the outcome variable they describe (such as the presence of a disease) may depend not only on the other variables but on the cluster context. Moreover, sharing of biomedical data is often hindered by patient confidentiality laws, and there is current interest in algorithms to generate synthetic tabular data from real data, for example via deep learning. We demonstrate a novel EM-based clustering algorithm, MMM ("Madras Mixture Model"), that outperforms standard algorithms in determining clusters in synthetic heterogeneous data, and recovers structure in real data. Based on this, we demonstrate a synthetic tabular data generation algorithm, MMMsynth, that pre-clusters the input data, and generates cluster-wise synthetic data assuming cluster-specific data distributions for the input columns. We benchmark this algorithm by testing the performance of standard ML algorithms when they are trained on synthetic data and tested on real published datasets. Our synthetic data generation algorithm outperforms other literature tabular-data generators, and approaches the performance of training purely with real data.

## Introduction

Tabular datasets, consisting of heterogeneous variable types (categorical, ordinal, numeric), are ubiquitous in data science and, in particular, in biomedical research. Such datasets may additionally be heterogeneous in rows: they may consist of a mix of different subtypes or categories, corresponding to hidden structures not originally measured or are part of the dataset (such as geographical location, socioeconomic class, genotype, etc).

For machine learning applications, often one variable (often called a "response" or "output" variable) is of clinical interest (such as presence or absence of a disease or disorder such as diabetes, birth weight of a fetus) and the goal is to train a model to predict it from the other measurable variables (often called "predictor" or "input" variables). Patient confidentiality often

rsidd120/MadrasMixtureModel} under the MIT licence. They are implemented in Julia.

**Funding:** We acknowledge funding from BIRAC grant bt/ki-data0404/06/18 (RS), and the IMSc Centre for Disease Modelling (ICDM) funded via an apex project at IMSc by the Department of Atomic Energy, Government of India (CK, RS). The funders had no role in the data collection, research, analysis, writing or submission of the manuscript.

**Competing interests:** The authors have declared that no competing interests exist.

restricts the ability to share such datasets freely, and several algorithms have been developed [1–3] to generate synthetic datasets that closely resemble real datasets and can be used to train ML models and shared freely.

Here we address both tasks, of clustering heterogeneous data, and of generating realistic synthetic datasets as measured by their performance in training models on them for ML prediction on the real data.

## Existing clustering methods

Several standard clustering algorithms exist for multidimensional data and are implemented in machine learning packages such as `scikit-learn` [4], `ClusterR` [5] and `Clustering.jl` [6]. These include agglomerative(hierarchical) clustering methods such as UPGMA, partitioning algorithms such as $K$-means that seek to optimize a particular distance metric across $K$ clusters, and mixture models such as Gaussian Mixture Models (GMMs) that assume that the data is drawn from a mixture of distributions and simultaneously learn the parameters of the distributions and the assignment of data points to distributions.

These methods assume an underlying appropriate distance metric (such as Euclidean distance) (agglomerative clustering, or $K$-means), or assume an underlying probability distribution for the data (GMMs) which is to be learned.

Many datasets consist of a mixture of binary (eg, gender) categorical (eg, ethnicity), ordinal (eg, number of children), and numerical data (eg, height, weight). Columns in such a tabular dataset may be correlated or interdependent.

## Overview of the MMM algorithm

Here we propose an algorithm, which we call the Madras Mixture Model (MMM), to cluster tabular data where the columns are assumed independently drawn from either categorical or real data. For each clustering, we optimize the likelihood of the total data being drawn from that clustering:

$$P(D|K) = \prod_{j=1}^{K} P(d : d \in j) \tag{1}$$

where $K$ is the number of clusters, and the right hand side is a product over clusters for the likelihood that the subset of rows $d$ that are assigned to cluster $j$ would be co-clustered (this is stated more precisely in Methods).

We assume that categorical variables are drawn from an unknown categorical distribution with a Dirichlet prior, and numeric variables are drawn from an unknown normal distribution with a normal-Gamma prior. These are the conjugate priors for the multinomial and normal distributions respectively: that is, they are analytically tractable and the posteriors have the same functional form as the priors, while the form is flexible enough to represent a variety of prior distributions.

Unlike with standard GMM algorithms, we do not attempt to estimate the parameters of the distributions, but integrate over them to obtain the likelihoods (see Methods). Our clustering approach is a variation of expectation maximization (EM) where, essentially, the M-step in the usual GMM algorithm is replaced with this integration.

To determine the true number of clusters, we use the marginal likelihood (ML) (sometimes called the Bayesian Occam's razor) [7]. While the Bayesian information criterion [8], is widely used as an approximation to the ML, its performance on our synthetic dataset benchmarks was inferior (see supplementary information).

The most accurate numerical calculation of the ML is using thermodynamic integration (TI) [9, 10], reviewed in Methods, and we provide an implementation using TI. This is computationally expensive since it involves sampling at multiple different "temperatures" and integrating. As a faster alternative, the ML is frequently estimated using the harmonic mean (HM) of samples [11], but this is known to give a biased estimate in practice [12]. We give an improved approximation, which we call HM$\beta$, involving a fictitious inverse temperature $\beta$, which, for suitable $\beta$, converges to the true value much faster than the HM on small datasets where the exact answer is calculable, and also converges rapidly to a fixed value on larger datasets. We demonstrate that HM$\beta$ with $\beta \approx 0.5$ produces results comparable to TI on our synthetic datasets.

## MMMSynth: Generating synthetic tabular data

Patient confidentiality is both an ethical requirement in general and a legal requirement in many jurisdictions. Clinical datasets may therefore only be shared if patients cannot be identified; "pseudonymization" (replacing real names with random IDs) is not enough in general since it may be possible to identify patients from other fields including age, clinical parameters, geographical information, etc.

Given the difficulty of sharing clinical datasets without violating patient confidentiality, there has been interest in using machine learning to generate synthetic data that mimics the charateristics of real tabular data. Several methods have been proposed [1–3]; these generally rely on deep learning or other sophisticated approaches.

We use MMM as a basis for a relatively straightforward synthetic tabular data generation algorithm, MMMsynth. Each cluster is replaced with a synthetic cluster which column-wise has the same statistical properties for the input variable, and whose output variable is estimated with a noisy linear function learned from the corresponding cluster in the true data. All these synthetic clusters are then pooled to generate a synthetic dataset.

We assess quality of synthetic data by the performance, in predicting on real data, of ML models trained on synthetic data. We demonstrate that this rather simple approach significantly outperforms other published methods CT-GAN and CGAN, and performs, on average, better than Gaussian Copula and TVAE. Our performance in many cases approaches the quality of prediction from training on real data.

# Materials and methods

## MMM: Clustering of heterogeneous data

Consider a tabular data set consisting of $L$ heterogeneous columns and $N$ rows. Each row of the set then consists of variables $x_i$, $i = 1, 2, \ldots, L$. Each $x_i$ can be binary, categorical, ordinal, integer, or real. We consider only categorical (including binary) or numeric data; ordinal or numeric integer data can be treated as either categorical or numeric depending on context. If there are missing data, they should first be interpolated or imputed via a suitable method. Various imputation methods are available in the literature: for example, mean imputation, nearest-neighbour imputation, multivariate imputation by chained equations (MICE) [13].

## Discrete data, dirichlet prior

For a categorical variable with $k$ values, the Dirichlet prior is $P_{(p)} \propto \prod_{i=1}^{k} p_i^{c_i-1}$. For binary variables ($k = 2$) this is called the beta prior. If we have already observed data $D$ consisting of $N$ observations, with each outcome $j$ occurring $N_j$ times, the posterior predictive for outcome $x =$

$i$ $(1 \leq i, j \leq k)$ is

$$P(x = i|D) = \frac{N_i + c_i}{N + C} \tag{2}$$

where $C = \sum_{i=1}^{k} c_i$.

## Continuous data, normal-gamma prior

For a continuous normally-distributed variable, we use a normal-gamma prior, as described in [14], with four hyperparameters, which we call $\mu_0, \beta_0, a_0, b_0$:

$$
\begin{aligned}
p(\mu, \lambda) \quad &= \mathcal{N}(\mu|\mu_0, (\beta_0\lambda)^{-1})\text{Gam}(\lambda|a_0, b_0) \\
&= \frac{(\beta_0\lambda)^{1/2}}{\sqrt{2\pi}} e^{-\frac{\beta_0\lambda}{2}(\mu - \mu_0)^2} \frac{1}{\Gamma(a_0)} b_0^{a_0} \lambda^{a_0-1} e^{-b_0\lambda}
\end{aligned}
\tag{3}
$$

$$
= \left(\frac{\beta_0}{2\pi}\right)^{1/2} \frac{b_0^{a_0}}{\Gamma(a_0)} \lambda^{a_0-1/2} \exp\left(-\frac{\lambda}{2}\left[\beta_0(\mu - \mu_0)^2 + 2b_0\right]\right). \tag{4}
$$

Here $\lambda$ is the inverse of the variance, $\lambda = \frac{1}{\sigma^2}$.

Given data $D$ consisting of $n$ items $x_i$, $i = 1 \ldots n$, the posterior is

$$
\begin{aligned}
p(\mu, \lambda|D) &= NG(\mu, \lambda|\mu_n, \beta_n, a_n, b_n) \\
&= \left(\frac{\beta_n}{2\pi}\right)^{1/2} \frac{b_n^{a_n}}{\Gamma(a_n)} \lambda^{a_n-1/2} \exp\left(-\frac{\lambda}{2}\left[\beta_n(\mu - \mu_n)^2 + 2b_n\right]\right)
\end{aligned}
\tag{5}
$$

where

$$\mu_n = \frac{\beta_0\mu_0 + n\bar{x}}{\beta_0 + n} \tag{6}$$

$$\beta_n = \beta_0 + n \tag{7}$$

$$a_n = a_0 + \frac{n}{2} \tag{8}$$

$$b_n = b_0 + \frac{1}{2}\sum_{i=1}^{n}(x_i - \bar{x})^2 + \frac{\beta_0 n(\bar{x} - \mu_0)^2}{2(\beta_0 + n)} \tag{9}$$

The posterior predictive, for seeing a single new data item $x$ given the previous data $D$, is [14]

$$p(x|D) = \pi^{-1/2} \frac{\Gamma\left(a_n + \frac{1}{2}\right)}{\Gamma(a_n)} \left(\frac{\Lambda}{2a_n}\right)^{\frac{1}{2}} \left(1 + \frac{\Lambda(x - \mu_n)^2}{2a_n}\right)^{-\left(a_n + \frac{1}{2}\right)} \tag{10}$$

where

$$\Lambda = \frac{a_n\beta_n}{b_n(\beta_n + 1)} \tag{11}$$

In log space:

$$\log p(x|D) = -0.5 \log \pi + \log \Gamma\left(a_n + \frac{1}{2}\right) - \log \Gamma(a_n) + 0.5(\log \Lambda - \log(2a_n))$$
$$-\left(a_n + \frac{1}{2}\right) \log\left(1 + \frac{\Lambda(x - \mu_n)^2}{2a_n}\right)$$

(12)

The marginal likelihood is

$$p(D) = \frac{\Gamma(a_n)}{\Gamma(a_0)} \frac{b_0^{a_0}}{b_n^{a_n}} \left(\frac{\beta_0}{\beta_n}\right)^{1/2} (2\pi)^{-n/2}$$

(13)

The posterior predictives for categorical (2) and normal (12) are used in the clustering algorithm described in the next section.

## Optimizing likelihood of a clustering by expectation maximization

Let the data $D$ consist of $N$ rows, so that $D_i$ is the $i$'th row. Let the model be denoted by $M_K$ where $K$ is the number of clusters. Each cluster has its own parameters of the categorical or normal distribution for each column which we call $\vec{\theta}$, with $\vec{\theta}_{\ell j}$ being the vector of parameters for column $\ell$ in cluster $j$. The vector has $k - 1$ independent components for a categorical distribution of $k$ categories, and 2 components for a normal distribution, which are all continuous. Another, discrete parameter for the model is the detailed cluster assignment of each row $D_i$ to each cluster $C_j$. This can be described by a vector $\vec{A}$ of length $N$, whose elements $A_i$ take values from 1 to $K$. A specific clustering is described by $\vec{\Theta} = \{\vec{\theta}, \vec{A}\}$.

Given $K$, we seek an optimal clustering (i.e, optimal choice of $\vec{A}$) by maximum likelihood. We write the likelihood being maximized is

$$\vec{A}^* = \underset{\vec{A}}{\operatorname{argmax}} \prod_{j=1}^{K} P(d_j)$$
$$= \underset{\vec{A}}{\operatorname{argmax}} \prod_{j=1}^{K} \int P(d_j|\vec{\theta}_j) P(\vec{\theta}_j) d\vec{\theta}_j.$$

(14)

Here $d_j$ is shorthand for $\{D_i | A_i = j\}$, that is, it is the set of rows from $D$ that belong to cluster $j$, and $\vec{\theta}_j$ is the set of parameters $\vec{\theta}$ specific to cluster $j$. The unknown parameters $\vec{\theta}_j$ are integrated over, separately for each cluster, and $P(\vec{\theta}_j)$ is a Dirichlet or normal-Gamma prior as appropriate. That is, we seek to find that assignment of individual rows to clusters, $\vec{A}$, such that the product of the likelihoods that the set of rows $D_j$ that have been assigned to the cluster $j$ are described by the same probabilistic model is maximized. There is an implicit product over columns, which are assumed independent.

In contrast to EM implementations of GMMs, we only seek to learn $\vec{A}$; we do not seek the parameters of the probabilistic model $\vec{\theta}$, but integrate over them.

An algorithm for clustering into $K$ clusters could be

1. Initialize with a random assignment of rows to clusters.

2. Calculate a score matrix $L_{ij}$. For each row $i$, this is the likelihood that $D_i$ belongs to cluster $j$ for all $j$ (excluding $i$ from its current cluster).

3. Assign each row $i$ to the cluster corresponding to that value of $j$ which maximises $L_{ij}$. (This is similar to the E-step in EM.)

4. Repeat from step 2 until no reassignments are made.

Instead of the M-step in EM, step 2 calculates likelihoods according to the posterior predictives for the categorical (2) and normal (10) distributions. The likelihood for a row is the product of the likelihoods over columns. We work with log likelihoods.

In our implementation, we iterate starting from one cluster. After optimizing each $K$ clustering, starting at $K = 1$, we use a heuristic to initialize $K + 1$ clusters: we pick the poorest-fitting $\frac{N}{K+1}$ rows (measured by their posterior predictive for the cluster that they are currently in) and move them to a new cluster, and then run the algorithm as above.

We can choose to either stop at a pre-defined $K$, or identify the optimal $K$ via marginal likelihood, as described below (the optimal $K$ could be 1).

## Identifying the correct $K$: Marginal likelihood

Eq 14 maximizes the likelihood of a clustering over $\vec{A}$, while marginalizing over $\vec{\theta}$. The correct $K$ in the Bayesian approach is the $K$ that maximises the marginal likelihood (ML) marginalized over *all* parameters including $\vec{A}$. In other words, while one can increase the likelihood in Eq 14 by splitting into more and more clusters, beyond a point this will lead to overfitting; the full ML penalizes this (an approach also called the Bayesian Occam razor [7]).

Unfortunately, exact calculation of the ML (marginalizing over $\vec{A}$) is impossible. There are several approaches using sampling; we review two below (harmonic mean [HM], and thermodynamic integration [TI]), before introducing our own, a variant of HM which we call HM$\beta$, which we show is more accurate than HM, and on our data, comparably accurate to TI while being faster.

**Arithmetic mean and harmonic mean.** A straightforward estimation of the marginal likelihood for $K$ clusters ($ML_K$) would be to sample uniformly from the parameter space for $\vec{A}$, and calculate the average likelihood over $M$ samples (there is an implicit marginalization over $\vec{\theta}$ throughout):

$$ML_K \equiv P(D|K) \approx \frac{1}{M} \sum_{m=1}^{M} P(D|K, \vec{A}_m) \tag{15}$$

This is the arithmetic mean (AM) estimate, and tends to be biased to lower likelihoods because the region of high-likelihood parameters is very small.

An alternative is to start from Bayes' theorem:

$$P(\vec{A}|D, K) = \frac{P(D|\vec{A}, K)P(\vec{A}|K)}{\sum_{\vec{A}'} P(D|\vec{A}', K)P(\vec{A}'|k)} \tag{16}$$

The denominator on the right is the marginal likelihood for $K$ clusters. Rearranging,

$$\frac{P(\vec{A}|K)}{ML_K} = \frac{P(\vec{A}|D, K)}{P(D|\vec{A}, K)} \tag{17}$$

and summing over $\vec{A}$ with $\sum_{\vec{A}} P(\vec{A}|K) = 1$,

$$\frac{1}{ML_K} = \sum_{\vec{A}} \frac{P(\vec{A}|D, K)}{P(D|\vec{A}, K)} \tag{18}$$

If we sample $\vec{A}$ from the distribution $P(\vec{A}|D, K)$, then for $M$ samples we have

$$ML_K \approx \left( \frac{1}{M} \sum_{m=1}^{M} \frac{1}{P(D|\vec{A}_m, K)} \right)^{-1}. \tag{19}$$

This is the "harmonic mean" (HM) approximation. Both the HM and the AM can be derived via different choices of an importance sampling distribution in a Metropolis-Hastings scheme [10]. The HM is known to be biased towards higher likelihoods in practice, oppositely to the AM.

**Thermodynamic integration.** Thermodynamic integration (TI), a technique borrowed from physics, was described in the statistical inference context by Gelman and Meng [9]. The following quick summary is adapted to our notation from Lartillot and Philippe [10].

Suppose one has an un-normalized density in parameter space, parametrized by $\beta$, $q_\beta(\vec{A})$. We can define a "partition function"

$$Z_\beta = \int q_\beta(\vec{A}) d\vec{A}. \tag{20}$$

In our case $\vec{A}$ is discrete, so the integral, here and below, should be interpreted as a sum. From this we get a normalized density

$$p_\beta(\vec{A}) = \frac{1}{Z_\beta} q_\beta(\vec{A}). \tag{21}$$

We then have

$$\begin{aligned}
\frac{\partial}{\partial \beta} \log Z_\beta &= \frac{1}{Z_\beta} \frac{\partial Z_\beta}{\partial \beta} \\
&= \frac{1}{Z_\beta} \frac{\partial}{\partial \beta} \int q_\beta(\vec{A}) d\vec{A} \\
&= \int \frac{1}{q_\beta(\vec{A})} \frac{\partial q_\beta(\vec{A})}{\partial \beta} \frac{q_\beta(\vec{A})}{Z_\beta} dA \\
&= E_\beta \left[ \frac{\partial \log q_\beta(\vec{A})}{\partial \beta} \right]
\end{aligned}$$

Defining $U(\theta) = \frac{\partial}{\partial \beta} \log q_\beta(\vec{A})$, and integrating from $\beta = 0$ to 1,

$$\log Z_1 - \log Z_0 = \int_0^1 E_\beta[U] d\beta. \tag{22}$$

Consider the particular choice

$$q_\beta(\vec{A}) = P(D|\vec{A}, K)^\beta P(\vec{A}|K) \tag{23}$$

where, as above, $K$ is the number of clusters. Then $q_0$ is the prior for $\vec{\theta}$, and $q_1$ is proportional to the posterior. Therefore $Z_0$ is 1 (since $q_0$ is normalized) and $Z_1$ is the marginal likelihood.

Substituting,

$$\log \mathrm{ML} = \int_0^1 E_\beta[\log P(D|\vec{A}, K)]d\beta. \tag{24}$$

The expectation $E_\beta$ is calculated by sampling at various $\beta$ and the integral is found by Simpson's rule.

**A faster approximation.** We return to the HM approximation (Eq 18). The problem is that the distribution $P(\vec{A}|D, K)$ is strongly peaked around the optimal parameters $\vec{A}$. To broaden the distribution we can introduce a fictitious inverse temperature $\beta$ (not the same as in TI), and write

$$\frac{1}{ML_K} = \sum_{\vec{A}} \frac{P(\vec{A}|D, K)^\beta P(\vec{A}|D, K)^{1-\beta}}{P(D|\vec{A}, K)} \tag{25}$$

But

$$P(\vec{A}|D, K) = \frac{P(D|\vec{A}, K)P(\vec{A}|K)}{ML_K} \tag{26}$$

and this gives

$$\frac{1}{ML_K} = \sum_{\vec{A}} P(\vec{A}|D, K)^\beta \left(\frac{P(\vec{A}|K)}{ML_K}\right)^{1-\beta} P(D|\vec{A}, K)^{-\beta} \tag{27}$$

and therefore

$$ML_K^{-\beta} = \sum_{\vec{A}} P(\vec{A}|D, K)^\beta P(\vec{A}|K)^{1-\beta} P(D|\vec{A}, K)^{-\beta} \tag{28}$$

We can evaluate $ML_K$ from this by sampling from

$$\tilde{P}(\vec{A}|D, K) \equiv P(\vec{A}|D, K)^\beta P(\vec{A}|K)^{1-\beta} \tag{29}$$

(in practice from $P(D|\vec{A}, K)^\beta$ which is proportional to this) rather than $P(\vec{A}|D, K)$. However, this distribution is not normalized. Let $\sum_{\vec{A}} \tilde{P}(\vec{A}|D, K) = \sum_{\vec{A}} P(\vec{A}|D, K)^\beta P(\vec{A}|K)^{1-\beta} = Z \neq 1$. If we take $M$ samples from $\tilde{P}(\vec{A}|D, K)$, then

$$ML_K^{-\beta} = \frac{1}{M} \sum_{\substack{m=1 \\ \text{samples from } \tilde{P}}}^{M} P(D|\vec{A}, K)^{-\beta} \times Z \tag{30}$$

$$\equiv \left\langle P(D|\vec{A}, K)^{-\beta} \right\rangle_{\tilde{P}} Z. \tag{31}$$

To estimate $Z$ we are forced to sample from the normalized distribution $P(\vec{A}|D,K)$:

$$Z = \sum_{\vec{A}} \tilde{P}(\vec{A}|D,K) = \sum_{\vec{A}} \frac{P(\vec{A}|D,K)^{\beta} P(\vec{A}|K)^{1-\beta}}{P(\vec{A}|D,K)} P(\vec{A}|D,K) \tag{32}$$

$$\equiv \left\langle \frac{P(\vec{A}|D,K)^{\beta} P(\vec{A}|K)^{1-\beta}}{P(\vec{A}|D,K)} \right\rangle_{P} \tag{33}$$

$$= \left\langle \frac{P(D|\vec{A},K)^{\beta-1} P(\vec{A}|K)^{\beta-1}}{ML_K^{\beta-1}} P(\vec{A}|K)^{1-\beta} \right\rangle_{P} \tag{34}$$

Substituting,

$$ML_K^{-\beta} = ML_K^{1-\beta} \langle P(D|\vec{A},K)^{-\beta} \rangle_{\tilde{P}} \langle P(D|\vec{A},K)^{\beta-1} \rangle_{P}$$

$$ML_K^{-1} = (D|\vec{A},K)^{-\beta} \rangle_{\tilde{P}} (D|\vec{A},K)^{\beta-1} \rangle_{P}$$

and finally

$$\log ML_K = -\log(D|\vec{A},K)^{-\beta} \rangle_{\tilde{P}} - \log(D|\vec{A},K)^{\beta-1} \rangle_{P} \tag{35}$$

where, again, $\langle \ldots \rangle_{\tilde{P}}$ means an average over $M$ samples from $P(D|\vec{A},K)^{\beta}$, and $\langle \ldots \rangle_{P}$ means an average over $M$ samples from $P(\vec{A}D,K)$. This expression reduces to the arithmetic mean if $\beta = 0$ and to the harmonic mean if $\beta = 1$. We refer to it as HM$\beta$.

We assess an optimal choice of $\beta$ using a small dataset of 20 rows where the marginalization can be carried out explicitly, by summing over all 1,048,574 possible cluster assignments to two clusters (Fig 1). The TI method converges quite quickly to the exact answer, and the HM$\beta$ for $\beta = 0.5$ also gives good results. On larger datasets too we found $\beta = 0.5$ an optimal choice (as in next subsection), though the exact value of the ML is not known.

**Synthetic data for benchmarking clustering.** We generated multiple datasets, each with 10 columns, 5000 rows, with five clusters each, in a 5:4:3:2:1 ratio, with varying parameters, as follows. For categorical data, columns in each cluster $j$ were sampled from vectors $\mathbf{v}_0 + \Delta \mathbf{v}_j$ where $\mathbf{v}_0$ was common to all clusters and $\mathbf{v}_j$ was specific to the $j$'th cluster, and $\Delta$ was varied from 0.5 to 4.5, with smaller values indicating greater similarity among clusters. Here, five columns were binary and five were 4-valued. For numeric data, the means in different clusters were separated by 1.0 and the standard deviations were separated by $\delta\sigma$, which varied from 0.5 to 4.5, with smaller $\delta\sigma$ indicating less dispersed, more distinct clusters (for small variance, clusters with different means have less overlap). We also simulated numeric data where the means in all clusters were the same and only the standard deviations differed by $\delta\sigma$. Finally, we generated mixed datasets, which included five numeric and five categorical (4-valued) columns, varying the parameter $\Delta$ as above and fixing $\delta\sigma = 5.0 - \Delta$, so that increasing $\Delta$ implies increasing similarity among all columns.

## Assessment of benchmarking

We used the adjusted Rand index (ARI) [15] to compare true and predicted clusters.

In supplementary information, we also show three other metrics: Normalized Clustering Accuracy, Fowlkes-Mallows Score, and Adjusted Mutual Information, as provided by the Genie-Clust [16] suite; results were similar. Some of these apply only when the true and predicted

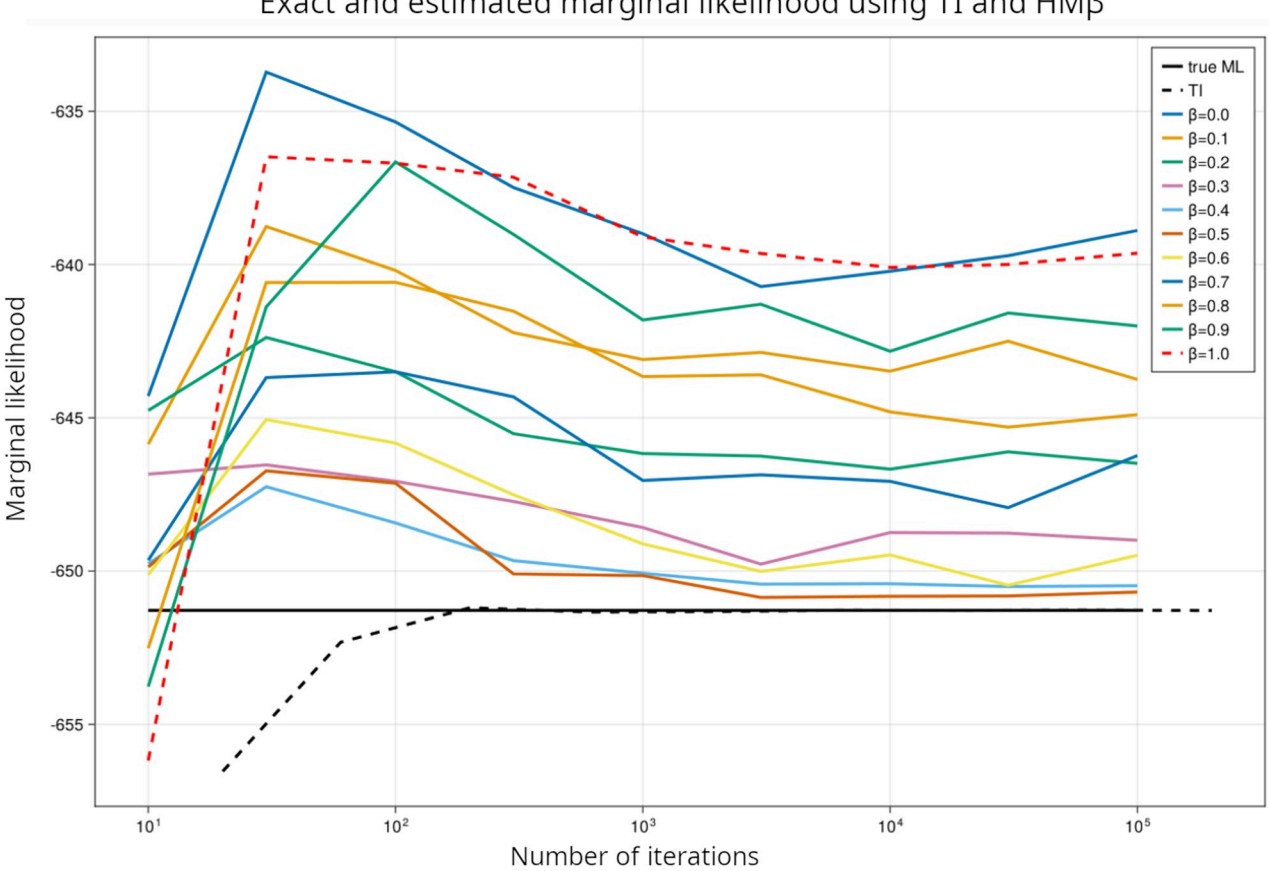

**Fig 1. For a dataset of 20 rows, the marginal likelihood can be calculated exactly (solid line); this is compared with TI and with HM*β* at various *β*.** *β* = 0.5 gives results close to the exact value.

numbers of clusters are the same. We also compare, for completeness, two internal clustering metrics (which do not depend on a known ground-truth clustering): silhouette score and Davies-Bouldin score. These require an internal distance metric to be defined for pairwise distances to be defined; we use the Euclidean distance, but MMM does not use a metric, but is based on a likelihood marginalized over hyperparameters, so these measures are to be treated with caution.

## MMMsynth: Generating synthetic data with MMM

**Synthetic data generation algorithm.**   MMMSynth uses MMM to pre-cluster an input dataset, excluding the output column. Each cluster is assumed, as in MMM, to consist of independent columns that are either categorical or numeric. The parameters of the corresponding multinomial or Gaussian distribution are fitted to each column in each cluster, and a new cluster of the same size is generated by sampling from these distributions. A linear model is fitted to the output column in each real cluster and used to generate the output column in the synthetic clusters. The synthetic clusters are finally combined to produce a full dataset of the same size as the original dataset.

For comparison we generated synthetic data using the methods available in literature. We used synthetic data vault [17] libraries in python to generate synthetic data using TVAE, Gaussian Copula, CTGAN and CGAN.

**Benchmarking MMMSynth-generated synthetic datasets.** To evaluate the similarity of the generated data of the real data, we trained machine learning models (logistic regression, random forest) on the synthetic data and evaluated their predictive performance on the real dataset. We also compared the performance of a model trained on the real dataset in predicting on the same dataset. We used six datasets from the UCI machine learning repository.

## Benchmarks: Real datasets used

We used the following datasets from the UCI Machine Learning Repository [18], some of which were obtained via the Kaggle platform:

- Abalone: predicting age of abalone from physical measurements, 8 predictors (input variables), 1323 rows, from UCI https://archive.ics.uci.edu/ml/datasets/abalone/ (note: original data had 4,177 rows and 28 output values, which are number of rings; for binary prediction we used only the rows with 9 or 10 rings, which were the most frequent, resulting in a dataset of 1323 rows of which 689 had 9 rings and 634 had 10 rings.)

- Heart failure prediction dataset: compiled from UCI Machine learning Repository, 11 predictors, 918 rows, from https://www.kaggle.com/datasets/rishidamarla/heart-disease-prediction

- Pima Indians diabetes: 8 predictors, 768 rows, https://www.kaggle.com/datasets/uciml/pima-indians-diabetes-database, source UCI

- Breast cancer Wisconsin (Diagnostic) dataset: 30 predictors, 569 rows, from https://archive.ics.uci.edu/ml/datasets/Breast+Cancer+Wisconsin+%28Diagnostic%29

- Maternal health risk data: 7 predictors, 676 rows, from https://www.kaggle.com/datasets/csafrit2/maternal-health-risk-data (note: original dataset had 1,014 rows with output values low, medium and high; for this task only 676 rows with output values low/high were retained.

- Stroke prediction dataset: 10 predictors, 4909 rows, of which 209 positive and 4700 negative, from https://www.kaggle.com/datasets/zzettrkalpakbal/full-filled-brain-stroke-dataset

- Connectionist Bench (Sonar, Mines vs. Rocks) dataset (60 predictors, 207 rows, from https://archive.ics.uci.edu/dataset/151/connectionist+bench+sonar+mines+vs+rocks

All of these have binary output variables. For clustering benchmarking, all were used (sonar was used as part of the ClustBench set, below). For synthetic data, the stroke dataset was omitted since it was highly unbalanced.

In addition, datasets from the UCI subdirectory of ClustBench [19] were used, many of which have multi-valued outputs. These consist of sonar (described above), as well as

- ecoli: 7 predictors, 335 rows, 8 categories

- glass: 9 predictiors, 213 rows, 6 categories

- ionosphere: 34 predictors, 350 rows, 2 categories

- statlog: 19 predictors, 2309 rows, 7 categories

- wdbc: 30 predictors, 568 rows, 2 categories

- wine: 13 predictors, 177 rows, 3 categories

- yeast: 8 predictors, 1483 rows, 10 categories

# Results

## Clustering algorithm performance

**Synthetic data: Purely categorical, purely normal, mixed.** We generate three kinds of synthetic datasets: purely categorical, purely numeric (normally distributed), and mixed, as described in Methods. Each dataset has five clusters. We vary a parameter ($\delta\sigma$ for numeric, $D$ for categorical), as described in Methods, to tune how similar the clusters are to each other. Large $\delta\sigma$ indicates a larger variance, and greater overlap among the "true" clusters; likewise small $D$ indicates that the categorical distributions of different clusters are closer. However, for numeric data, if clusters have the same mean, we expect that larger $\delta\sigma$ would improve the ability to separate them.

Fig 2 shows the results for normalized accuracy: for purely normally distributed data MMM performs comparably with Gaussian mixture models, for purely categorical data we greatly outperform all methods, with only $K$-means with one-hot encoding coming close, and for mixed normal+categorical data, too, our performance is superior to other methods. We run MMM in three modes: telling it the true cluster size, or using the TI and HM$\beta$ methods (with $\beta = 0.5$). All other methods are told the correct cluster size.

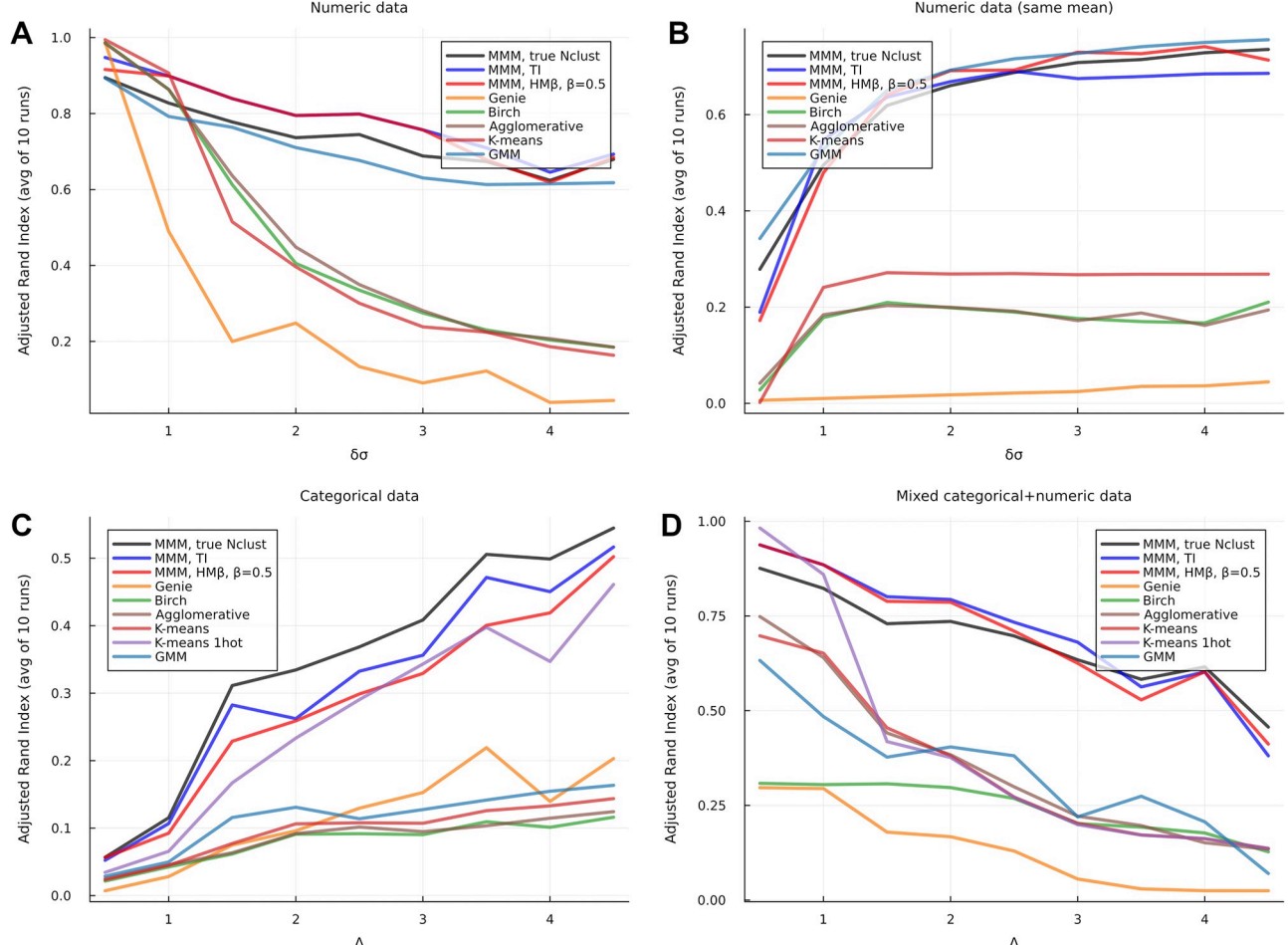

**Fig 2.** Clustering of four kinds of synthetic datasets: (A) purely numeric (normally distributed, differing means and variances), (B) purely numeric (normally distributed, same mean but differing variances), (C) purely categorical, and (D) mixed.

For numeric data with differing means, and for categorical data, MMM outperforms all other methods, but GMM comes close. For numeric data with the same mean and differing variances, GMM slightly outperforms MMM. For categorical data, K-means with 1-hot encoding comes close in performance to MMM. In dealing with mixed categorical+numeric data, MMM clearly outperforms all methods.

It is worth noting that the slopes of the ARI curves are opposite in Fig 2 A/D (differing mean) and B (uniform mean). This is because, when the means are different, a small variance makes the clusters more distinct and a large variance increases the overlap, but when the means are the same, the clusters can only be distinguished by their variance, so a larger $\delta\sigma$ helps.

MMM with the true Nclust does not always appear to perform best by this metric, but the performance with true Nclust, TI and HM$\beta$ are comparable in all cases.

**Predicting the true number of clusters.** We generated mixed categorical + numeric data, similar to above with 5000 rows per file, but divided into equal-sized clusters of 2 to 10 clusters. Fig 3 shows the performance of TI and HM$\beta$ in predicting the true number of clusters; both perform comparably, while the Bayesian information criterion (BIC) performs poorly here, likely accounting for its poor performance in the previous benchmark (not shown).

**Real data.** We consider the UCI abalone, breast, diabetes, heart, MHRD and stroke datasets described in Methods, which have binary outcome variables, and also eight datasets from the UCI machine learning database which are included in ClustBench [19], which have output labels of varying number from 2 to 10. These are intended as tests of classification, not clustering, tasks. Nevertheless, clustering these datasets on input variables (excluding the outcome variable) often shows significant overlap with the clustering according to the output variable, as shown in Fig 4. This suggests that we are recovering real underlying structure in the data. We compare various other methods, all of which were run with the correct known number of clusters, while MMM was run with ("MMM, true Nclust") and without ("MMM" telling it the true number of clusters.

The MMM results are with TI; the results with HM$\beta$ are similar and not shown for space reason.

Compared to the synthetic data results, several methods outperform MMM (even with fixed Nclust) on individual datasets, and MMM performs poorly on yeast, while all methods perform poorly on a few others. However, in most cases, MMM is among the better performers; in some cases the best performers, and in other cases inferior to another method, but no method is consistently superior.

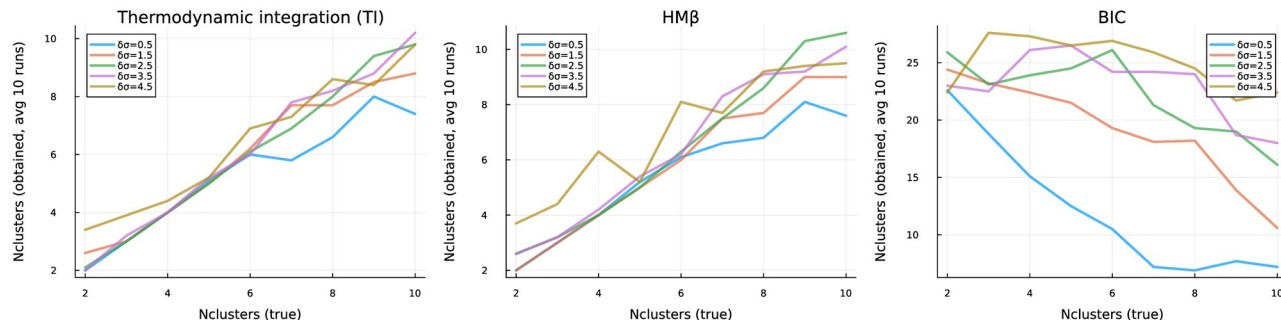

**Fig 3. Optimal number of clusters obtained by TI, HM$\beta$ and Bayesian Information Criterion (BIC), on mixed categorical+numeric data with true cluster number ranging from 2 to 10.** TI and HM$\beta$ show comparably good results while, in our data, BIC is mostly unable to predict the true number of clusters.

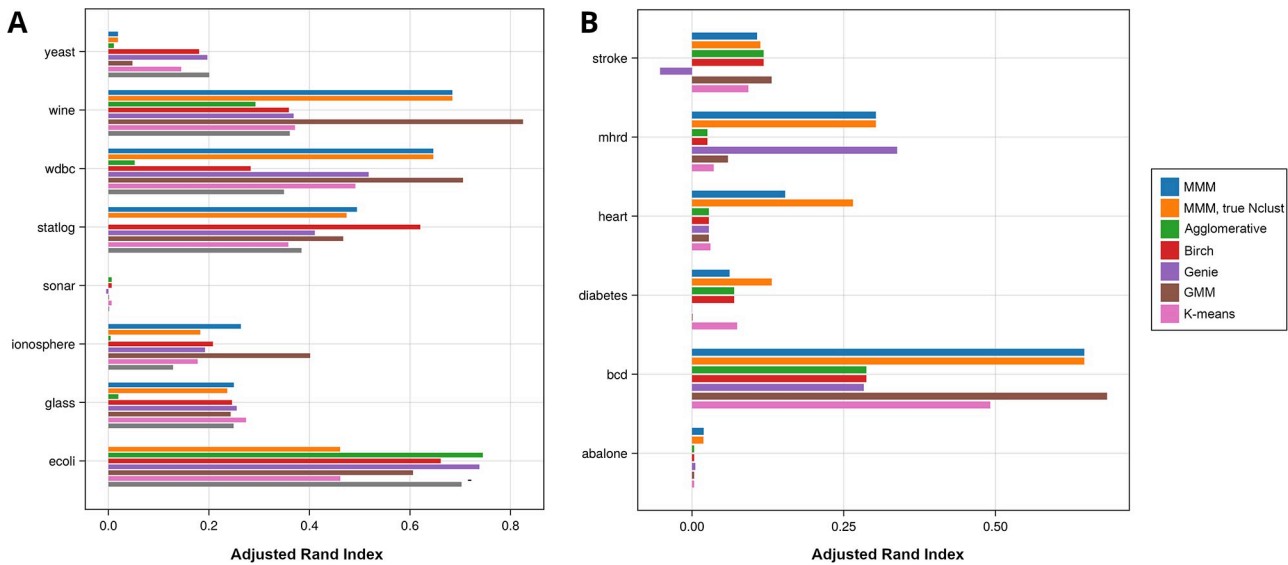

**Fig 4.** Performance of MMM, using the TI criterion, and "MMM, true Nclust" where it is told the true number of clusters versus other programs, on eight datasets from ClustBench (A) and six datasets from UCI (B). Performance is reported using Adjusted Rand Index.

When not told the true number of clusters, in some cases MMM refuses to cluster the dataset at all (as per the TI calculation of marginal likelihood, a single cluster is more likely than two). This is the case with the ecoli dataset.

**MMMSynth: ML performance on synthetic data.** We use the six kaggle/UCI datasets (Abalone, Breast Cancer, Diabetes, Heart, MHRD and Sonar) benchmarked above to generate synthetic data, as described in Methods, train ML algorithms (logistic regression, random forest) on these, and measure the predictive performance on the real data. Fig 5 show the results. Also tested are Triplet-based Variational Autoencoder (TVAE), Gaussian Copula (GC), Conditional Generative Adversarial Network (CTGAN), Copula Generative Adversarial Network (CGAN). AUC are averaged over 20 runs in each case. In all datasets, we show performance comparable to training on real data. TVAE and GC also perform well on most datasets, while CGAN and CTGAN show poorer performance. All programs were run with default parameters.

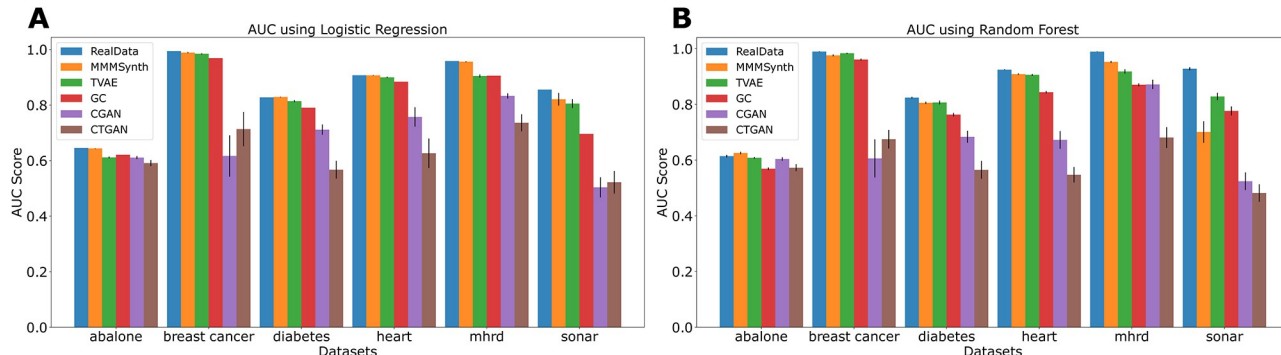

**Fig 5.** Logistic regression (A) and random forest (B) models were trained on the real data and on synthetic data generated using MMMSynth, TVAE, GC, CGAN and CTGAN and their predictive performance evaluated on the real datasets. The AUC (area under ROC curve, averaged over 20 runs) is shown for each method and each dataset, and errorbars are shown too.

## Discussion

We present a clustering algorithm, MMM, that clusters heterogeneous data consisting of categorical and numeric columns. We demonstrate good performance on a variety of publicly available datasets and on our synthetic data benchmarks. Speed optimizations will be explored in future work.

Currently the columns are assumed to be independent, but it will be a straightforward exercise to use a multivariate Gaussian to describe the numeric columns. This too will be explored in future. Despite this, our performance is comparable to and sometimes better than scikit-learn's GMM implementation on the real-world UCI and clustbench data presented here. In some benchmarks, other methods (Birch for statlog, Agglomerative and Genie for ecoli) clearly outperform both us and GMM. We emphasise again that the labels in these benchmarks are not ground truth clustering labels, but output labels in classification datasets. While (as commonly recognized), there is no universal "best" clustering method, we perform well across a wide variety of synthetic and real-data benchmarks, and further improvements are possible. in the future.

We further use MMM as a basis for a synthetic data generation algorithm, MMMSynth. We demonstrate that MMMSynth generates synthetic data that closely resembles real datasets. Our method performs better than current synthetic data generation algorithms in the literature (TVAE, GC, CGAN and CTGAN), though the gap with TVAE in particular is small. Notably, all these methods explicitly assume and model correlations between input columns: CGAN and GC use copula functions to capture correlations between variables, while all methods other than GC (CGAN, TVAE and CTGAN) employ deep learning using all columns. In contrast, we first cluster the data and then assume that, within each cluster, columns are uncorrelated. Our method requires modest computational resources and no deep learning, and we expect it will improve with improvements to the underlying MMM clustering algorithm.

Our approach indirectly accounts for some correlations: for example, if two binary columns are correlated (1 tends to occur with 1, and 0 with 0), we would be likely to cluster the 1's together and 0's together. It will also account for multimodal numeric columns since these would be better represented as a sum of Gaussians. In tests on synthetically-generated non-normally-distributed numeric data (for example, Gamma-distributed with long tails), MMM breaks the data into multiple clusters, suggesting an attempt to approximate the Gamma distribution as a sum of Gaussians. This will be explored in a future work. Nevertheless, we do not see such a proliferation of clusters when running on real datasets.

Currently, missing data needs to be imputed via another algorithm such as nearest-neighbour or MICE imputation. MMMSynth could be the basis for an alternative imputation algorithm in the future.

## Supporting information

**S1 File. This supporting information file explores the use of metrics other than Adjusted Rand Index; specifically Normalized Clustering Accuracy, Adjusted Mutual Information, Fowlkes-Mallows, as well as two intrinsic measures: Davies-Bouldin and silhouette score.** (PDF)

## Acknowledgments

This work grew from a project on personalized clinical predictions, for which we acknowledge discussion and collaboration with Gautam Menon, Uma Ram, Ponnusamy Saravanan, and particularly Leelavati Narlikar with whom we extensively discussed this work and whose

insights were invaluable. We also acknowledge useful discussions with Durga Parkhi on synthetic data generation.

## Author Contributions

**Conceptualization:** Chandrani Kumari, Rahul Siddharthan.

**Data curation:** Chandrani Kumari, Rahul Siddharthan.

**Formal analysis:** Chandrani Kumari, Rahul Siddharthan.

**Funding acquisition:** Rahul Siddharthan.

**Investigation:** Chandrani Kumari, Rahul Siddharthan.

**Methodology:** Chandrani Kumari, Rahul Siddharthan.

**Project administration:** Rahul Siddharthan.

**Resources:** Rahul Siddharthan.

**Software:** Chandrani Kumari, Rahul Siddharthan.

**Supervision:** Rahul Siddharthan.

**Validation:** Chandrani Kumari, Rahul Siddharthan.

**Visualization:** Chandrani Kumari, Rahul Siddharthan.

**Writing – original draft:** Chandrani Kumari, Rahul Siddharthan.

**Writing – review & editing:** Chandrani Kumari, Rahul Siddharthan.

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
