## [Decision Letter · Decision Letter 0]

16 Jan 2024

PONE-D-23-34859MMM and MMMSynth: Clustering of heterogeneous tabular data, and synthetic data generationPLOS ONE

Dear Dr. Siddharthan,

Thank you for submitting your manuscript to PLOS ONE. After careful consideration, we feel that it has merit but does not fully meet PLOS ONE’s publication criteria as it currently stands. Therefore, we invite you to submit a revised version of the manuscript that addresses the points raised during the review process.

We look forward to receiving your revised manuscript.

Kind regards,

Mohd Amril Nurman Mohd Nazir

Academic Editor

PLOS ONE

3. Thank you for uploading your study's underlying data set. Unfortunately, the repository you have noted in your Data Availability statement does not qualify as an acceptable data repository according to PLOS's standards.

4. We notice that there is a MIT license on your data. We would encourage you to consider using a license that is no more restrictive than CC BY, in line with PLOS’ recommendation on licensing (http://journals.plos.org/plosone/s/licenses-and-copyright).

Additional Editor Comments:

Please make all the revisions as suggested by reviewers.

Reviewers' comments:

Reviewer's Responses to Questions

**Comments to the Author**

1. Is the manuscript technically sound, and do the data support the conclusions?

Reviewer #1: Partly

Reviewer #2: Yes

Reviewer #3: Yes

2. Has the statistical analysis been performed appropriately and rigorously? 

Reviewer #1: Yes

Reviewer #2: Yes

Reviewer #3: Yes

3. Have the authors made all data underlying the findings in their manuscript fully available?

Reviewer #1: Yes

Reviewer #2: Yes

Reviewer #3: Yes

4. Is the manuscript presented in an intelligible fashion and written in standard English?

Reviewer #1: Yes

Reviewer #2: Yes

Reviewer #3: Yes

5. Review Comments to the Author

Reviewer #1: The present manuscript by Kumari and co-authors introduces a new clustering method (named MMM) and a new method for the generation of synthetic data (named MMMSynth). The manuscript presents the mathematical formulations for their methods introducing the reader to the underlying concept of the proposed method. The performance (from the perspective of two metrics) of the proposed clustering method has been evaluated against other five clustering algorithms on three synthetically generated datasets with varying values of standard deviations (decreasing and increasing the difficulty of clustering). Moreover, the authors have validated their method (in three variants, with two approximations for the number of clusters and the true number of clusters) is in comparison to five other clustering algorithms (that have been given the true number of clusters) on multiple datasets that have binary (seven datasets) and multiclass (eight datasets) labels from the perspective of a performance metric.

Major points:

Section 3.1.1: As you mention, the increase of the δσ on the x-axis will result in more similar clusters, nevertheless, all methods are becoming worse for Figure 2 panels A and D with higher values of δσ (that indicate more similar clusters). This is not discussed in the manuscript. The performance metric choice changing across different data sets (Figure 2 panel A vs panels B,C,D) is again not discussed. I would recommend either keeping ARI across all as the normalized accuracy appears just in Figure 2A or extending all analyses to include both metrics.

Section 2.6: Multiple perspectives are offered from multiple metrics, such as Adjusted Mutual Information/V-Measure, Purity, Fowlkes-Mallows. And even from internal metrics such as Silhouette, Davies-Bouldin, Calinski-Harabasz. This section has two sentences at this moment, I would recommend extending and discussing the choice of metrics as the normalized accuracy is used in a single analysis.

Overall, the figures are hardly discussed/interpreted in the manuscript, they are only referenced. I would recommend extending the manuscript with an interpretation of the results. Figures 2 and 3 could be improved, as the same methods are used, you could have the same legends (the addition of sklearn in the names of Figure 3 is not relevant as they are exactly the same methods as in figure 2). From Figure 2 to Figure 3, the name "MMM, true nClust" has been changed into "MMM (fixed)", making the manuscript harder to read. As a question, why is Figure 3 lacking the MMM, HMβ option? This is not discussed either. Figure 3 could have “ARI” added as the x-axis label.

Regarding the claims in section 4. Discussion, that the proposed methods outperform others:

• In Figure 3 panel A, GMM actually shows a better performance than the proposed method (even when the proposed method is given the true number of clusters) for all datasets.

• The results from Figure 4 do not exactly show this. TVAE might even outperform the proposed method on average for RandomForest and GC is a close contender as well. By averaging the performances from Figure 4, panel A, it seems that TVAE has overall a higher value. For panel B, the proposed method will probably have a slightly higher value but not by much, I would estimate around 3%. I would recommend more analyses, as at this point, the proposed generation method does not seem to bring a considerable improvement in comparison to TVAE.

Minor points:

Section 3.1: As far as I understand δσ will result in more dispersed clusters as it changes the standard deviation of the cluster. Thus, by similar do you mean that they will have more similar densities?

Section 2.8: You specify that the first list of datasets have binary output variables. But the abalone dataset has an integer value in the number of rings which is not binary. At least that is the case of the abalone dataset shown in the link.

Section 2.3: The mathematical formulations seem sound, although I have a question, shouldn’t it be μn in equation (5)?

Section 2.3: In the first paragraph, there is a period added after reference [13], while the sentence continues after it.

Section 2.7.1: In the second paragraph, “available” written as “availbale”.

Section 4: The GC acronym has been defined above, yet it is not used in the 4. Discussion section

I would recommend extending the github code. It is hard to use with the only output available as a file of labels. A plot (2D/3D through PCA) of the result of clustering a synthetically generated dataset in comparison to another clustering method (such as K-Means), would be a helpful addition. At this point, validation of the code requires additional code to be written by others.

Reviewer #2: MMM and MMMSynth: Clustering of heterogeneous tabular data, and synthetic data

generation

The manuscript’ MMM and MMMSynth: Clustering of heterogeneous tabular data, and synthetic data

generation’ is recommended for minor correction. The manuscript requires to address in the following aspects:

1. Introduction section is not followed the standard. This should be rewritten.

2. Section 2.3 is not clear, not identified why the equations are require for this method?

3. In section 4, equation 14 is your own expression, If not please use the proper reference. Please use references for other section when use equations.

4. TI method, abbreviation?

5. The manuscript needs to proofread by Authorised affiliations.

6. Conclusion needs to rewrite has a lack of summarise of findings.

7. You may include similar additional references: i) Review on the Evaluation and Development of Artificial Intelligence for COVID-19 Containment https://doi.org/10.3390/s23010527; ii) Bio-activity prediction of drug candidate compounds targeting SARS-Cov-2 using machine learning approaches, https://doi.org/10.1371/journal.pone.0288053 ; iii) Evaluating the Brexit and COVID-19’s influence on the UK economy: A data analysis https://doi.org/10.1371/journal.pone.0287342;

8. I recommend to use the comparison with existing models/methods.

Reviewer #3: Summary:

In this work, authors proposed novel methods to cluster heterogeneous tabular data and generate synthetic datasets. Based on the likelihood for clustering heterogeneous datasets, the proposed method employed EM algorithm to derive accurate clustering results. This work would be interesting for related research field and the manuscript is well-written as well. However, there are few comments to enhance the quality of the manuscript. Please check out the following comments:

Major:

1. It would be recommended to introduce related works (or literature review) in the introduction section. Although the introduction section well describes the background of the proposed work, if the related works or publications are introduced, it helps potential readers in making a follow-up study (or extension) of the proposed work.

2. Brief description of the proposed work may not appropriate in the introduction section.

“Here we propose an algorithm, which we call the Madras Mixture Model (MMM),……..Eq (1). ……….. . Our performance in many cases approaches the quality of prediction from training on real data.”

It would be recommended to make a new section such as “overview” to describe the compact explanation of the proposed work.

3. In the section 2.1: “If there are missing data, they should first be interpolated or imputed via a suitable method”

It would be good to provide more detailed description of interpolation or imputation methods if there are missing data. If the proposed method cannot handle the datasets including missing information, please clearly discuss its limitation in the discussion section.

4. In the sections 2.2 & 2,3: Please provide more explanation why you employ “Dirichlet prior” for the discrete data and “normal-gamma prior” for continuous data.

5. In the figure 3 A: Overall performance of MMM is not comparable to scikitlearn_gmm for most cases and scikitlearn_birch for yeast, statlog, ecoli. It would be good, if you can describe the reasons (or acceptable explanations) for inferior performance. Additionally, there is no results of MMM for ecoli. Do you have any reason for skipping the specific result? If it is, please clearly describe why it cannot the ARI for ecoli.

6. It would be recommended to present a brief result on how the proposed method can accurately predict the true number of clusters. For instance, given different datasets, you can set x-axis as the true number of clusters and y-axis as the estimated number of clusters through the proposed method.

6. PLOS authors have the option to publish the peer review history of their article (what does this mean?). If published, this will include your full peer review and any attached files.

Reviewer #1: **Yes: **Eugen-Richard Ardelean

Reviewer #2: No

Reviewer #3: No

---

## [Author Response · Author response to Decision Letter 0]

8 Feb 2024

We have uploaded a "response to reviewers file" separately. Below is a cut-paste but the file that we uploaded may be more readable. 

---

We thank all three reviewers for their positive reviews and useful feedback on this manuscript and are confident that our revision will answer their concerns. 

Reviewer #1: 

The present manuscript by Kumari and co-authors introduces a new clustering method (named MMM) and a new method for the generation of synthetic data (named MMMSynth). The manuscript presents the mathematical formulations for their methods introducing the reader to the underlying concept of the proposed method. The performance (from the perspective of two metrics) of the proposed clustering method has been evaluated against other five clustering algorithms on three synthetically generated datasets with varying values of standard deviations (decreasing and increasing the difficulty of clustering). Moreover, the authors have validated their method (in three variants, with two approximations for the number of clusters and the true number of clusters) is in comparison to five other clustering algorithms (that have been given the true number of clusters) on multiple datasets that have binary (seven datasets) and multiclass (eight datasets) labels from the perspective of a performance metric.

------

We thank the reviewer for this clear and accurate summary of our work. 

-----

Major points:

Section 3.1.1: As you mention, the increase of the δσ on the x-axis will result in more similar clusters, nevertheless, all methods are becoming worse for Figure 2 panels A and D with higher values of δσ (that indicate more similar clusters). This is not discussed in the manuscript. The performance metric choice changing across different data sets (Figure 2 panel A vs panels B,C,D) is again not discussed. I would recommend either keeping ARI across all as the normalized accuracy appears just in Figure 2A or extending all analyses to include both metrics.

The use of normalized accuracy on figure 2A was an oversight, and we have now used ARI in all cases, with normalized accuracy and other metrics in supporting data. The results are similar. We have now clarified why the ARI (or normalized accuracy) increases in figure 2B but decreases in figures 2A and 2D with increasing δσ. An increasing δσ in figures 2A and 2D indicates more similar clusters, which makes it harder to cluster accurately, resulting in decreasing ARI. But in figure 2B, the means of the clusters are the same, and the only distinguishing feature is the variance; an increasing δσ makes it easier to distinguish the clusters in that case. (page 11, lines 329-333; updated figure 2, page 12)

Section 2.6: Multiple perspectives are offered from multiple metrics, such as Adjusted Mutual Information/V-Measure, Purity, Fowlkes-Mallows. And even from internal metrics such as Silhouette, Davies-Bouldin, Calinski-Harabasz. This section has two sentences at this moment, I would recommend extending and discussing the choice of metrics as the normalized accuracy is used in a single analysis.

-----

We compare multiple metrics now (adjusted MI, Fowlkes-Mallows, Normalized Accuracy), in supplementary information (new), but the results are broadly similar. It should be noted that internal metrics require distance metrics to be defined between rows, which is not the case in our algorithm: our metric is the likelihood of the data given a cluster assignment, marginalized over the hyperparameters for each assignment via integrating with the appropriate conjugate prior. Nevertheless, we report silhouette and Davies-Bouldin scores in supplementary information, using Euclidean distance as metric.

----

Overall, the figures are hardly discussed/interpreted in the manuscript, they are only referenced. I would recommend extending the manuscript with an interpretation of the results. Figures 2 and 3 could be improved, as the same methods are used, you could have the same legends (the addition of sklearn in the names of Figure 3 is not relevant as they are exactly the same methods as in figure 2). From Figure 2 to Figure 3, the name "MMM, true nClust" has been changed into "MMM (fixed)", making the manuscript harder to read. As a question, why is Figure 3 lacking the MMM, HMβ option? This is not discussed either. Figure 3 could have “ARI” added as the x-axis label.

---

We have addressed all of these points. We now discuss the figures further in the results section. We have updated the captions of figure 3 to match figure 2. We omitted the HMβ criterion from figure 3 to save space and because the results were similar to TI; we mention this now (page 12, lines 353-354). We have added the x-axis caption to figure 

---

3.

Regarding the claims in section 4. Discussion, that the proposed methods outperform others:

• In Figure 3 panel A, GMM actually shows a better performance than the proposed method (even when the proposed method is given the true number of clusters) for all datasets.

This is not quite true: in figure 3A, we do marginally better than GMM for glass and statlog (though other methods do even better in those cases). And we did say that our performance is "comparable to and sometimes better than" GMM on these sets (despite the independence assumption). But yes, we should not overstate the improvement in our method and have rewritten the claims in Discussion to have a more balanced perspective (page 14, lines 383-388). As is frequently stated in the literature, there is no universal "best" method for clustering. 

• The results from Figure 4 do not exactly show this. TVAE might even outperform the proposed method on average for RandomForest and GC is a close contender as well. By averaging the performances from Figure 4, panel A, it seems that TVAE has overall a higher value. For panel B, the proposed method will probably have a slightly higher value but not by much, I would estimate around 3%. I would recommend more analyses, as at this point, the proposed generation method does not seem to bring a considerable improvement in comparison to TVAE.

---

This again is true: our improvement over TVAE is slight. However, we have re-done this analysis by averaging AUC over 20 runs and including error bars for all methods. In the updated figure (new figure 5) it appears that MMMsynth slightly outperforms TVAE on average in all cases. We are again careful in this revision not to overstate this , but we note the simplicity of MMMsynth, in contrast to the other methods which all use deep learning and require significant computational power (page 15, lines 393-399).

---

Minor points:

Section 3.1: As far as I understand δσ will result in more dispersed clusters as it changes the standard deviation of the cluster. Thus, by similar do you mean that they will have more similar densities?

We have clarified this (page 9, lines 231-232). It was badly worded. Indeed, δσ will result in more dispersed “true” clusters, which will therefore overlap and be harder to separate by a clustering program. But if the means are the same (figure 2B), the clusters overlap anyway, and in this case larger δσ will improve separability.

Section 2.8: You specify that the first list of datasets have binary output variables. But the abalone dataset has an integer value in the number of rings which is not binary. At least that is the case of the abalone dataset shown in the link.

---

We apologize for this oversight. We had preprocessed the abalone data to consider only 9 and 10 rings, but had omitted to mention this in Methods. We have clarified this now, and a similar clarification for maternal health risk data (page 10, lines 272-275 and 285-287). 

---

Section 2.3: The mathematical formulations seem sound, although I have a question, shouldn’t it be μn in equation (5)?

---

It should indeed be μn, we thank the referee for this correction which has been done.

---

Section 2.3: In the first paragraph, there is a period added after reference [13], while the sentence continues after it.

---

Fixed.

---

Section 2.7.1: In the second paragraph, “available” written as “availbale”.

---

Fixed.

---

Section 4: The GC acronym has been defined above, yet it is not used in the 4. Discussion section

---

We now use the acronym in Discussion

---

---

I would recommend extending the github code. It is hard to use with the only output available as a file of labels. A plot (2D/3D through PCA) of the result of clustering a synthetically generated dataset in comparison to another clustering method (such as K-Means), would be a helpful addition. At this point, validation of the code requires additional code to be written by others.

---

The point is valid. We have now included a demo notebook for using the code via a jupyter notebook, including a visualization example. We plan to make the code more "julia-like" and usable as a module in the future, and also include a scikit-learn-compatible interface for python. 

---

Reviewer #2: 

MMM and MMMSynth: Clustering of heterogeneous tabular data, and synthetic data generation. 

The manuscript’ MMM and MMMSynth: Clustering of heterogeneous tabular data, and synthetic data generation’ is recommended for minor correction. The manuscript requires to address in the following aspects:

1. Introduction section is not followed the standard. This should be rewritten.

---

We have improved the presentation in the introduction, also in response to reviewer 3. In particular we have included subheadings and additional literature review.

---

2. Section 2.3 is not clear, not identified why the equations are required for this method?

---

We have added a mention that eq 12 (and 2) are used in the clustering algorithm described in the next section. (page 4, lines 114-115)

---

3. In section 4, equation 14 is your own expression, If not please use the proper reference. Please use references for other section when use equations.

---

It is our own equation. We have tried to make this clear (page 5, line 128).

---

4. TI method, abbreviation?

---

This (Thermodynamic Integration) is defined in the title of a subsection in Methods, where it is first introduced. (page 7, lines 182-183)

---

5. The manuscript needs to proofread by Authorised affiliations.

---

We have authorised a colleague with excellent English to read it and hope there are no remaining errors

---

 6. Conclusion needs to rewrite has a lack of summarise of findings.

---

We have rewritten the discussion to emphasise the results more.

---

7. You may include similar additional references: i) Review on the Evaluation and Development of Artificial Intelligence for COVID-19 Containment https://doi.org/10.3390/s23010527; ii) Bio-activity prediction of drug candidate compounds targeting SARS-Cov-2 using machine learning approaches, https://doi.org/10.1371/journal.pone.0288053 ; iii) Evaluating the Brexit and COVID-19’s influence on the UK economy: A data analysis https://doi.org/10.1371/journal.pone.0287342;

---

These papers (which we note are all from the same group) do not seem to be relevant to our work. 

---

8. I recommend to use the comparison with existing models/methods.

---

This was already done, both for clustering (multiple methods) and for synthetic data generation (GC, CTGAN, CGAN, TVAE). 

---

Reviewer #3: 

Summary:

In this work, authors proposed novel methods to cluster heterogeneous tabular data and generate synthetic datasets. Based on the likelihood for clustering heterogeneous datasets, the proposed method employed an EM algorithm to derive accurate clustering results. This work would be interesting for related research fields and the manuscript is well-written as well. However, there are few comments to enhance the quality of the manuscript. Please check out the following comments:

Major:

1. It would be recommended to introduce related works (or literature review) in the introduction section. Although the introduction section well describes the background of the proposed work, if the related works or publications are introduced, it helps potential readers in making a follow-up study (or extension) of the proposed work.

---

We thank the reviewer for the positive comments. We have now included a brief review of both clustering and tabular data generation in the introduction. (page 2 and 3)

---

2. Brief description of the proposed work may not appropriate in the introduction section.

“Here we propose an algorithm, which we call the Madras Mixture Model (MMM),……..Eq (1). ……….. . Our performance in many cases approaches the quality of prediction from training on real data.”

It would be recommended to make a new section such as “overview” to describe the compact explanation of the proposed work.

---

While the PLOS One "manuscript organization" style does not include such an overview section, we have reorganized the introduction into subsectoins, including overview subsections for MMM and MMMsynth, to describe our work.

---

3. In the section 2.1: “If there are missing data, they should first be interpolated or imputed via a suitable method”

It would be good to provide a more detailed description of interpolation or imputation methods if there are missing data. If the proposed method cannot handle the datasets including missing information, please clearly discuss its limitation in the discussion section.

---

We have now given examples of imputation methods, with reference (page 3, lines 94-96). We have also included a mention in Discussion (page 15, lines 409-411).

---

4. In the sections 2.2 & 2,3: Please provide more explanation why you employ “Dirichlet prior” for the discrete data and “normal-gamma prior” for continuous data.

---

These are the conjugate priors for the categorical and normal distributions respectively. We have said a few words about this now (page 2, lines 43-46).

---

5. In the figure 3 A: Overall performance of MMM is not comparable to scikitlearn_gmm for most cases and scikitlearn_birch for yeast, statlog, ecoli. It would be good, if you can describe the reasons (or acceptable explanations) for inferior performance. Additionally, there is no results of MMM for ecoli. Do you have any reason for skipping the specific result? If it is, please clearly describe why it cannot the ARI for ecoli.

---

We have clarified this now. MMM with the TI criterion does not find significant clusters for ecoli, and returns the entire dataset as a single cluster, resulting in an ARI of 0. (page 14, lines 360-362)

---

6. It would be recommended to present a brief result on how the proposed method can accurately predict the true number of clusters. For instance, given different datasets, you can set x-axis as the true number of clusters and y-axis as the estimated number of clusters through the proposed method.

---

We have now compared the three methods (TI, HMβ, and BIC) in predicting the true number of clusters, in synthetic data where the true number of clusters varies from 1 to 10, in new figure 3. TI and HMβ both perform well while BIC performs poorly. (page 12, lines 336-341; new figure 3)

---

## [Decision Letter · Decision Letter 1]

1 Apr 2024

MMM and MMMSynth: Clustering of heterogeneous tabular data, and synthetic data generation

PONE-D-23-34859R1

Dear Dr. Siddharthan,

We’re pleased to inform you that your manuscript has been judged scientifically suitable for publication and will be formally accepted for publication once it meets all outstanding technical requirements.

Kind regards,

Zeyar Aung

Academic Editor

PLOS ONE

Additional Editor Comments (optional):

Reviewers' comments:

Reviewer's Responses to Questions

**Comments to the Author**

1. If the authors have adequately addressed your comments raised in a previous round of review and you feel that this manuscript is now acceptable for publication, you may indicate that here to bypass the “Comments to the Author” section, enter your conflict of interest statement in the “Confidential to Editor” section, and submit your "Accept" recommendation.

Reviewer #1: All comments have been addressed

Reviewer #3: All comments have been addressed

2. Is the manuscript technically sound, and do the data support the conclusions?

Reviewer #1: Yes

Reviewer #3: Yes

3. Has the statistical analysis been performed appropriately and rigorously? 

Reviewer #1: Yes

Reviewer #3: Yes

4. Have the authors made all data underlying the findings in their manuscript fully available?

Reviewer #1: Yes

Reviewer #3: Yes

5. Is the manuscript presented in an intelligible fashion and written in standard English?

Reviewer #1: Yes

Reviewer #3: Yes

6. Review Comments to the Author

Reviewer #1: My comments have been addressed. Nevertheless, I would like to mention the following:

1. If you check the ClustBench documentation, the datasets you have chosen (ecoli, yeast, wine, … - for example in Fig4A) are actually from UCI as well:

“A selection of 8 high-dimensional datasets available through the UCI (University of California, Irvine) Machine Learning Repository [12]. Some of them were considered for benchmark purposes in, amongst others, [30]. They are also listed in the sipu battery. However, their original purpose is for testing classification, not clustering algorithms. Most clustering algorithms find them problematic; due to their being high-dimensional, it is difficult to verify the sensibleness of the reference labels.”

2. The writing style could be improved

Reviewer #3: In this revision, all raised issues are well addressed and properly discussed. One minor recommendation is that, if you give a tutorial page (or instruction) in the GitHub, it would increase the usability of the proposed method.

7. PLOS authors have the option to publish the peer review history of their article (what does this mean?). If published, this will include your full peer review and any attached files.

Reviewer #1: **Yes: **Eugen-Richard Ardelean

Reviewer #3: No

---

## [Editor Report · Acceptance letter]

5 Apr 2024

PONE-D-23-34859R1 

PLOS ONE

Dear Dr. Siddharthan, 

I'm pleased to inform you that your manuscript has been deemed suitable for publication in PLOS ONE. Congratulations! Your manuscript is now being handed over to our production team.

Kind regards, 

on behalf of

Dr. Zeyar Aung 

Academic Editor

PLOS ONE